# *In Vitro* Measurements of Cellular Forces and their Importance in the Lung—From the Sub- to the Multicellular Scale

**DOI:** 10.3390/life11070691

**Published:** 2021-07-14

**Authors:** Peter Kolb, Annika Schundner, Manfred Frick, Kay-E. Gottschalk

**Affiliations:** 1Institute of Experimental Physics, Ulm University, 89069 Ulm, Germany; peter.kolb@uni-ulm.de; 2Institute of General Physiology, Ulm University, 89069 Ulm, Germany; annika.schundner@uni-ulm.de

**Keywords:** lung, force sensing, mechanotransduction

## Abstract

Throughout life, the body is subjected to various mechanical forces on the organ, tissue, and cellular level. Mechanical stimuli are essential for organ development and function. One organ whose function depends on the tightly connected interplay between mechanical cell properties, biochemical signaling, and external forces is the lung. However, altered mechanical properties or excessive mechanical forces can also drive the onset and progression of severe pulmonary diseases. Characterizing the mechanical properties and forces that affect cell and tissue function is therefore necessary for understanding physiological and pathophysiological mechanisms. In recent years, multiple methods have been developed for cellular force measurements at multiple length scales, from subcellular forces to measuring the collective behavior of heterogeneous cellular networks. In this short review, we give a brief overview of the mechanical forces at play on the cellular level in the lung. We then focus on the technological aspects of measuring cellular forces at many length scales. We describe tools with a subcellular resolution and elaborate measurement techniques for collective multicellular units. Many of the technologies described are by no means restricted to lung research and have already been applied successfully to cells from various other tissues. However, integrating the knowledge gained from these multi-scale measurements in a unifying framework is still a major future challenge.

## 1. Introduction

Throughout life, the lung is exposed to mechanical forces. Mechanical forces play an important role in lung development, maturation, and function and can contribute to the pathogenesis of lung disease. Understanding lung mechanics has played a central role in understanding lung biology and respiratory physiology [1]. The mechanical forces in the lung include stretching, compressing, and shear stresses, as well as forces derived from surface tension and substrate rigidity. These are generated, sensed, and responded to at different scales—from subcellular structures to the tissue or organ level. At the organ scale, global lung mechanics have largely been inferred from pressure, volume, and flow relationships, as well as, very recently, digital image correlation techniques [2,3]. In recent years, novel imaging technologies, including intravital microscopy [4,5,6], fast synchrotron radiation CT imaging [7,8,9], and magnetic resonance elastography [10,11], have added information on tissue deformations on the macro- and micro-scale level and added to a better understanding of the local mechanical properties in airways and alveoli. In addition, a plethora of *in vitro* assays have identified the mechanical properties of lung cells, extracellular matrix, and lung tissues and highlighted the mechanisms of the mechanotransduction in the lung [12].

Based on these findings, many studies inferred detailed cellular responses to specific mechanical forces. However, the actual quantification of these forces is often difficult. A detailed understanding and characterization of the mechanical forces affecting the cellular responses is necessary to differentiate between physiological and pathophysiological stimuli and to better translate the findings from *in vitro* models to the actual *in vivo* situation [13]. In this short review, we give a brief overview of the mechanical forces at play on the cellular level in the lung. We focus on the technological aspects of measuring these from subcellular forces to measuring the collective behavior of cellular networks. We also want to emphasize that many of the technologies described are by no means restricted to lung research and have already been applied successfully to cells from various other tissues.

## 2. Forces on Lung Cells and Tissue

Mechanical forces generate stress at the subcellular and cellular level. Stresses deform the original shape of a body or cell and are expressed in units of Pa (N/m^2^) [14]. The resulting deformation can be quantified by a relative length change or a relative angle change and is called strain. Strain has no unit. If the deformability is completely reversible, it is called elastic deformation. The ratio of stress and strain is then the elastic or Young’s modulus. Stiff materials have a high Young’s modulus, while soft materials have a low modulus. Bone typically has a Young’s modulus on the order of GPa, while healthy lung tissue has a Young’s of 1–5 kPa. When the lung becomes scarred and fibrotic, the Young’s modulus increases manyfold and reaches values of up to 100 kPa [15].

Cell strain or stretch in the lung has received considerable attention. Many *in vitro* and *in vivo* experiments highlighted profound effects of cell stretch on cell function. Different cell types, including epithelial, endothelial, and mesenchymal cells, are subjected to (continuous cyclic) cell stretch and compression in the lung. Part of the stretching forces in the alveolus also result from the surface tension at the air-liquid interphase, which has important effects on the alveolar micro-architecture [15]. In the alveolus, stretch stimulates cell proliferation and differentiation during fetal lung development [16,17,18,19,20,21], stimulates the secretion of pulmonary surfactant from alveolar type 2 cells [6,22,23,24,25,26,27,28,29], affects alveolar barrier function [30,31] and repair mechanisms [32], among other effects. However, stretch above the physiological range can also lead to alveolar cell and tissue damage [6,33,34], increased pro-inflammatory responses [35,36,37], or impaired wound healing [38,39], resulting in alveolar instability, endothelial leakage, alveolar edema, surfactant dysfunction, or tissue remodeling. These pathological changes are associated with the pathogenesis of severe pulmonary diseases, including acute respiratory distress syndrome (ARDS), ventilator induced lung injury (VILI) [40,41,42,43], and fibrosis [44,45]. Similar to the alveolus, stretching or compressing airway cells elicits multiple physiological and pathological responses. These include establishing respiration at birth and maintaining normal breathing in adults [46], airway constriction [46,47,48], pathological airway remodeling [49,50,51], the generation of pro-inflammatory responses [52,53], and the development of airway hyperresponsiveness [54]. Again, the dose (magnitude of stress) determines whether the outcomes are beneficial or detrimental.

Shear stress also plays a significant role within the lung. Shear stress is the force exerted per cross-sectional area and leads to the deformation of cells. It results from the movement of a fluid along a boundary. Interactions between fluid (e.g., airway surface liquid) and a boundary (e.g., epithelial cells), together with friction between fluid molecules, leads to a velocity gradient within the fluid and to a transfer of force onto the boundary. In the lung, shear stress, in particular, is induced by airflow at the air–liquid interphase of the airways. Shear stress exerted by the airflow is proportional to the changes in airway pressure [55]. Physiological levels of shear stress impact the regulation of the airway surface liquid [56,57], mucus secretion [58], and overall muco-ciliary clearance and enhance the epithelial barrier function [59]. Elevated shear stress on the epithelial cell layer might play a part in tissue damage, inflammation, and perhaps even airway remodeling [60]. In the alveoli, a special form of shear stress, interfacial stress, stimulates the secretion of pulmonary surfactant and affects the gene expression in alveolar type 2 cells [61,62]. Apart from shear stress transmitted by airflow in the airways, shear stress within blood vessels and the microvasculature is significant and impacts endothelial function [63], but it also contributes to the pathogenesis of pulmonary arterial hypertension or proinflammatory signaling cascades [64,65].

In recent years, forces exerted by the rigidity (stiffness) of the extracellular matrix (ECM) have received special attention. The stiffness of the ECM is defined by its elastic or Young’s modulus, which is the ratio of stress and strain and characterizes the ECM´s resistance to deformation in response to applied force. The healthy lung tissue has a Young’s modulus of 1–5 kPa which increases manyfold and reaches values of up to 100 kPa when the lung becomes scarred and fibrotic [66]. It has been found that ECM stiffness regulates human airway smooth muscle contraction [67]. However, ECM-dependent forces have received particular attention due to their contribution to the pathogenesis of fibrotic diseases [68,69,70,71]. Aberrant increases in matrix stiffness result in the activation of lung fibroblasts, excessive ECM deposition, and fibrotic remodeling of the lung tissue [66], whereas reducing stiffness can partially reverse a stiffness-induced myofibroblast phenotype [72]. In addition to its rigidity, the topography of the ECM, i.e., the architecture, geometry, size, and organization of the matrix network, also exerts stresses in cells, leading to changes in the cell shape or differentiation [73].

Apart from the obvious impact of mechanical forces on tissue-embedded cells that are directly exposed to stretch, shear, or ECM-dependent mechanical signals via cell–cell, cell–matrix, or cell–flow interactions, recent studies also highlighted the effects of mechanical forces on the cells of the immune system [74]. In the lung, it has been demonstrated that cyclical hydrostatic pressure initiated an inflammatory response by immune cells [75].

Excellent reviews have summarized the multiple responses to the different mechanical forces in the lung in detail [1,70,73,76,77,78,79]. Substantial progress has also been achieved in delineating how cells sense external physical cues and translate them into a cellular response. While force-sensing occurs in or on the plasma membrane, forces can reach deep in the cell interior and to the nucleus [80,81,82]. Specific mechano-transducing elements on the cell surface, including cell adhesion protein complexes, primary cilia, G protein-coupled receptors, and mechanically gated ion channels [29,83,84,85,86], have been identified in lung cells, and progress has been achieved in deciphering signaling pathways that ultimately result in physiological or pathophysiological cellular responses [87,88]. Part of this progress has been achieved through the development of accurate technologies for the precise application and investigation of the complex intracellular, intercellular, and cell-matrix forces affecting lung development, function, and pathophysiological processes.

## 3. Cellular Force Measurement

In recent years, a wide range of methods have been developed to measure the mechanical properties of cells, forces generated by cells, and the response of cells to external mechanical stimuli. These are divided into active and passive methods (Figure 1) [89]. When using active methods, a force is applied to a cell or cell surface, and the response of the cell to its deformation is measured. In contrast, passive methods do not apply mechanical stimuli to cells but observe how cellular forces alter the cell environment. While many methods can be upscaled for multicellular samples, the special mechanics of cell aggregates and tissue also allows for further measurement methods [90]. To characterize multicellular mechanics, a continuum approach can be used, where the local mechanics are represented as the average of local volumes containing several cells [91,92].

### 3.1. Active Cellular Force Probing

Different approaches can be applied to exert mechanical stimuli on cells or tissues and to measure their response. They can be divided into methods that deform only small parts of a cell and those that deform whole cells or tissue.

The measurement of the mechanics at the subcellular scale requires techniques using probes in the (sub)micrometer range. The aspiration of the cell membrane into a micropipette can be utilized to deform parts of a cell surface (Figure 2a) [93,94]. Hereby, stress is applied locally by a pressure gradient between the inside and outside of the micropipette. This stress leads to cellular strain: the cell membrane deforms and gets sucked into the pipette. How much strain is induced by the applied stress is dependent on the mechanical properties of the cell. Therefore, monitoring the cell edge inside the pipette and knowing the applied suction pressure allows for calculating values like surface tension to gain insight into cell cortex mechanics, the influence of motor proteins, the viscosity of the cytoplasm of soft cells, like chondrocytes, and the elastic modulus of stiffer cells, like endothelial cells [95]. For calculating elastic properties, it is important that the strain induced by sucking in parts of the membrane does not lead to a permanent deformation but is completely reversible. Micropipette aspiration can also be used to study the mechanical properties of tissue [96,97]. Similar to single cells, the aspiration of tissue into larger pipette tips allows for the determination of the viscosity and the elastic modulus [98,99].

In addition to inducing strain by pulling on cells, cellular mechanics can also be studied by pushing probes into the surface of cells and measuring their response. Here again, the local mechanical properties can be monitored. Pushing locally onto the cell surface allows for measuring the cortex tension (convoluted with the mechanical properties of the cytoskeleton). This can be achieved using an Atomic Force Microscope (AFM). Starting as a method for imaging the surfaces of inorganic materials [100], AFM has found its way into the field of biomechanics [101,102,103] and allows for gaining information about the mechanical properties [104,105,106] and adhesion [107] of cells. Based on the idea of measuring a force by the displacement of a spring, an AFM consists of a soft cantilever with a small tip (Figure 2b). Using the tip, forces in the pN to nN range can be exerted on the cell surface at sub-micrometer length scales, and the response is directly probed under physiological conditions [108]. To measure the displacement of the tip, a laser that reflects off the back of the cantilever onto a photodiode is used [109,110]. This allows for measurement in different environments, including under physiological conditions [108]. Besides being used as a tool for studying topography, AFM allows for the exerting of forces in the pN to nN range at sub-micrometer length scales and probe response in order to gain information about mechanical properties [104,105,106] and adhesion [107]. Similar to micropipette measurements, elastic and viscous properties can be determined [111,112,113].

In order to measure the relative elastic modulus of cells, a cantilever with a spherical tip can be used. Radmacher et al. [104] showed that using the Hertz model, the relationship between the force $F_{C}$ on the cantilever and the Young’s modulus *E* of the cell is given by:(1)$$F_{C}=\frac{4}{3}\frac{E}{\left(1-v\right)}\sqrt{R}\;\delta^{\frac{3}{2}}$$
where *R* is the tip radius, *υ* is the Poisson ratio, and δ is the indentation. By measuring the vertical position (distance) and the deflection of the cantilever at a known cantilever stiffness, the force distance (FD) curves can be recorded at different positions on a cell. In Figure 3a, an AFM height image of a mouse fibroblast is shown. The measured FD curves for four different positions marked in Figure 3a are shown in Figure 3b. Using Equation (1), those curves can be fitted to calculate the Young’s modulus at each position.

AFM has been used to study the mechanics of different cell types [114,115,116] and has helped in understanding the underlying processes in various diseases, including cancer [117,118].

Besides AFM, other techniques, including cell pokers apparatuses [119,120,121,122] and micro-needles [123,124,125,126], have been used to exert indentation forces on cell surfaces and measure cellular responses. At the tissue-scale, instruments with larger tips, like optical fibers [127] or micro-indenters [128], can be utilized. Instead of using small tips, microplates can be used to probe the mechanical properties of cells [129,130] or cell aggregates [131]. Centrifugation of cell aggregates and, subsequently, drop shape analysis allow for the calculation of surface tension at the macro-scale [132,133]. Since these techniques are perfectly compatible with a variety of classical imaging methods, like widefield, phase contrast, and fluorescence microscopy, applying strain locally to single cells or multicellular units can be leveraged for analyzing biochemical, strain-dependent pathways with appropriate labelling strategies.

Optical and magnetic tweezers allow for the possibility to exert and measure cellular forces in the pN range, without the use of cantilevers. Since the discovery that a laser beam can trap microscopic objects [134], optical tweezers have proven their utility in a wide range of biological applications [135,136,137]. The intensity gradient in a focused laser beam allows particles with a higher refractive index than the surrounding medium to be trapped near its center (focus). By calibrating the trap, the necessary forces to push particles out of the trap center can be measured. While often being used as a tool to study single molecules, such as kinesin [138] or myosin [139], optical tweezers also provide insight into the mechanics of larger cellular structures. Cytoskeletal transport can be investigated by introducing beads into cells [140], and the mechanical properties of the cell membrane can be investigated by attaching beads to the cell surface [141]. Similar to AFM techniques, a bead can be trapped and pushed into the cell surface to measure the elastic response [142]. Additionally, whole cells can be stretched in a setup with two opposing laser beams (Figure 2c) [143], which allows for the determination of whole cell elasticity [144,145]. This has already been used for the mechanical phenotyping of cells to detect malignant [146,147] or mechanical changes of mesenchymal stem cells after doubling [148]. Optical traps have also been used to study the cell-cell-junction stiffness in cell aggregates [149]. In addition to the optical trap, magnetic tweezers can be used for active micro-rheology [150]. Similar to a macroscopic rheometer, intracellular ferromagnetic microbeads are twisted in a magnetic field [151] to determine the viscoelastic properties of the cytoplasm (Figure 2d) [152]. Beads attached to a cell membrane have also been used to study cellular mechanotransduction [153]. At the tissue level, large-scale methods, like magnetomotive optical coherence elastography, use similar magnetic effects [154].

Lasers are also used to disrupt cell–cell junctions [155] or cut through tissue [156,157]. Laser ablation of cellular structures disrupts force transmission and leads to a movement of the surrounding tissue. From the movement of the cell edges, the cell-junction tension can be inferred. At the macro-scale, a scalpel can also be used for dissection, instead of a laser [158].

To investigate the cellular mechanics of cells and tissue that are under constant shear stress, as in blood vessels or lungs, microfluidic devices [159,160,161,162] and mechanical stretchers [163,164,165,166] can be used. In these devices, the cell usually experiences a shear stress, resulting from the strained (stretched) substrate or the moving fluid. This shear stress leads to mechanical strain of the cell. Monitoring the biochemical consequences of the cellular strain induced by these techniques allows for the investigation of biomechanical signaling pathways. Altering the substrate stiffness in mechanical stretchers adds a further dimension to these investigations. With this, it is possible to apply constant or cyclic forces to single cells or layers and observe cellular responses, like the activation of mechano-sensitive piezo1 ion channels [29].

### 3.2. Passive Cellular Force Mapping

Passive cellular force mapping techniques use special molecules, particles, or substrates, instead of actively applying forces to cells. Monitoring the cells´ adaptive behavior and shape changes is then used to calculate the cellular mechanics.

Observing the cellular forces of adherent cells on elastic substrates requires the tracking of substrate displacement. Initially, this was achieved by monitoring the wrinkles caused by cellular contraction on soft silicone substrates [167]. However, the methodology and resolution of traction force microscopy (TFM) have been significantly improved over the years [168]. The first improvements came with the introduction of substrates with tunable compliance and the precise measurement of substrate stiffness [169,170]. Since it is difficult to calculate forces from the shape of wrinkles, beads were introduced to track the substrate deformation (Figure 4a) [171,172]. With the advancement in micropatterning, substrates could be structured, and beads could be arranged in defined arrays. This enabled the study of cellular forces and their relation to the focal adhesion assembly [173]. Instead of using flat substrates, it is also possible to seed cells on micropillars (Figure 4b) [174] for the accurate tuning of the substrate stiffness [175] and precise calculation of the cellular forces exerted on individual pillars [176]. The relation between the force *F* applied by a cell to one pillar and the deflection δ is given by:(2)$$F=\frac{3EI\delta }{L^{3}}$$
where *E* is the Young’s modulus, *I* is the moment of inertia, and *L* is the pillar length [174]. Figure 5a depicts a smooth muscle cell sitting on top of such a micropillar array (from Tan et al. [174]). The fluorescently stained fibronectin allows for the calculation of the force applied by the cells. Additional staining of focal adhesion protein vinculin (Figure 5b, from [174]) revealed a relation between the generated force and the area of focal adhesion staining per post (Figure 5c). Since tracking of the micropillar displacement can be challenging, improvements have been made by introducing nanoparticles for large-area and high-precision tracking [177]. Besides seeding cells on a deformable substrate, it has also been shown that subcellular forces created by individual adhesive contacts can be measured by allowing adherent cells to attach to the lever of a specifically micromachined device [178]. TFM can further be used to study cell aggregates and tissue [179], and techniques have been developed that can track forces of cells embedded in gels, which allow for the study of stresses in three-dimensional systems mimicking physiological conditions [180]. TFM substrates can also be incorporated into many other devices [181], including microfluidic channels [182,183] or cell stretchers [184], and have been used to study force generation during morphogenesis [185]. While it is difficult to track forces inside tissues due to the lack of a substrate, droplets have been introduced into tissues, and their deformation has been used to determine cell-generated forces [186].

Förster-Resonance-Energy-Transfer (FRET) has been applied to track cellular forces. Initially, FRET-based sensors were developed as molecular tension sensors [187,188]. Since then, they have been used to measure force transmission in cells [189] at cell–substrate [190] and at cell–cell [191] contacts. In FRET, two fluorophores in close proximity can undergo an efficient energy transfer when the donor is excited. The transfer rate is dependent on the distance between the two molecules and allows for a precise measurement of distance changes. This can be used to create artificial molecular tension sensors for TFM (Figure 4c) [188], but it also allows for tension measurements on molecules participating in the mechanotransduction inside cells by attaching donor and acceptor fluorophores to specific molecules [192,193]. Here, the strain is directly measured: the FRET signal only changes when the distance between fluorophores changes. Hence, it can be directly measured how externally applied stress leads to internal cellular strain. Göhring et al. [194] designed a FRET-based molecular force sensor that connects a glass-supported lipid bilayer to a T-cell receptor (Figure 5d). The present forces lead to a stretching of the sensor, which moves the acceptor and donor fluorophore further apart from each other and reduces the FRET efficiency (Figure 5e).

Another tool to determine mechanical properties in cells is passive micro-rheology. In contrast to its active counterpart, which uses magnetic tweezers to apply forces (see above), passive micro-rheology uses internalized microparticles that move through Brownian motion (Figure 4d). In these measurements, no strain or stress is actively induced in the cells. However, if they are combined with devices like microfluidic chambers, the effect of stress or strain on cellular mechanics can be probed. This allows for the study of the dynamic forces inside the cytoplasm and determination of the physical properties of the cytoskeletal network [195,196,197,198,199]. Passive micro-rheology can also be conducted with cells on different substrates, including micropatterned ones [200].

While most methods use some kind of probe or sensor to test cellular responses, novel techniques are able to directly infer forces via label-free imaging. Polarization microscopy allows for image stress anisotropy, which leads to a differences in refractive indices, called birefringence [201]. This can be used to study the substrate deformation caused by cellular traction forces, but also to determine the stress anisotropy in cells [202]. Brillouin microscopy is another emerging, non-invasive imaging technique that allows for the determination of viscoelastic properties in three dimensions [203]. It is based on the effect that acoustic phonons have due to density fluctuations, which can interact with light and lead to spectral changes caused by scattering. These spectral shifts reveal the states of photons, which can be used to determine viscoelastic properties [204]. A major advantage of this technology is that it can be applied over various scales, ranging from the sub- (intracellular) to the multicellular/tissue scale.

In cell aggregates and tissue, stresses can also be inferred from observing cell shapes and curvatures at cell–cell contacts [205]. Known as force inference, this method has improved over the years due to computational advances [206,207,208]. It has especially found an application in the study of morphogenetic movements [209].

## 4. Outlook

The lung is one of the organs where the mechanical properties of cells and extracellular matrix are tightly connected to the physiological function. The well-orchestrated interplay between different cell types in the lung, together with the three-dimensional, optimized architecture [210] poses a great challenge in terms of correlating quantitative measurements of cellular and extracellular stress and strain and its relation to physiology. The above-described techniques are fundamental for understanding the intricate interplay between mechanical load, cellular deformation, matrix remodeling, and the physiological function of the lung. So far, the techniques suffer from a low throughput and technical complexity, and they are often restricted to individual cells or smaller cellular units. Apart from further improving active and passive force measurement technologies, it will be of great importance to not only understand the effect of stress and strain on individual cells, but also to correlate these data with multi-cellular assays that also involve the extracellular matrix [211,212]. In particular, the measurement of the strain and forces in organs or whole organisms is urgently needed [213]. Many additional organs, like the bladder, heart, skin, or bone, are constantly exposed to a variety of forces. These forces are strongly coupled for function and are related to the stiffness of the tissues. Understanding and leveraging the mechanobiological response of tissue therefore bears great potential in fields ranging from stem cell reprogramming [214] to tissue engineering [215].

## Figures and Tables

**Figure 1 life-11-00691-f001:**
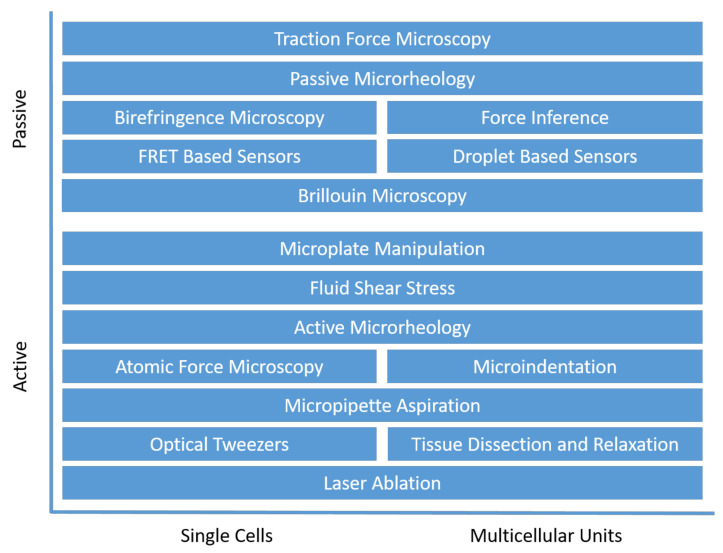
Overview of different methods for measuring cellular forces, organized into active and passive methods and grouped by the suitability for single-cell or multicellular unit studies.

**Figure 2 life-11-00691-f002:**
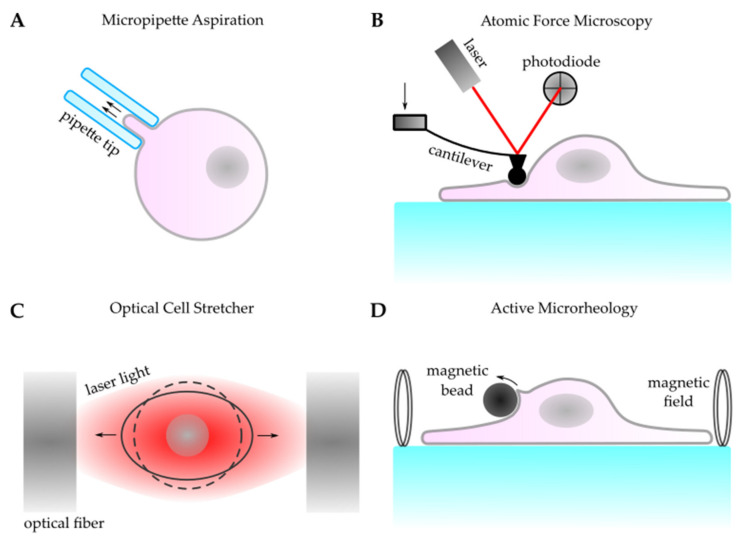
Schematics of active methods for measuring cellular forces and mechanical properties. (**A**) In micropipette aspiration, a force is applied to a cell through suction with a micropipette tip. Observation of the cell edge at known pressures allows for the determination of viscoelastic properties. (**B**) In Atomic Force Microscopy, a cantilever with a small tip is pressed into a cell, and its displacement is measured through a laser that reflects off its back onto a photodiode. This allows for mapping the topography and measuring the viscoelastic properties and molecular adhesion. (**C**) An optical cell stretcher consists of two opposing laser beams that create an electromagnetic field, which exerts forces that lead to the stretching of whole cells. Cell deformation with a calibrated system allows for the calculation of cellular elasticity. (**D**) Magnetic microbeads attached to a cell surface exert torque due to a magnetic field. Similar to a macroscopic rheometer, these can be used to determine the viscoelastic properties of the cell.

**Figure 3 life-11-00691-f003:**
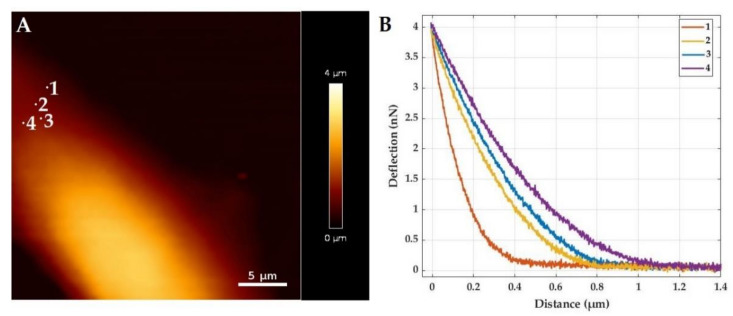
AFM data showing the variation in stiffness of a mouse fibroblast. (**A**) Measured height of a mouse fibroblast, showing the surface morphology. (**B**) Force distance curves taken from four different positions shown in (**A**).

**Figure 4 life-11-00691-f004:**
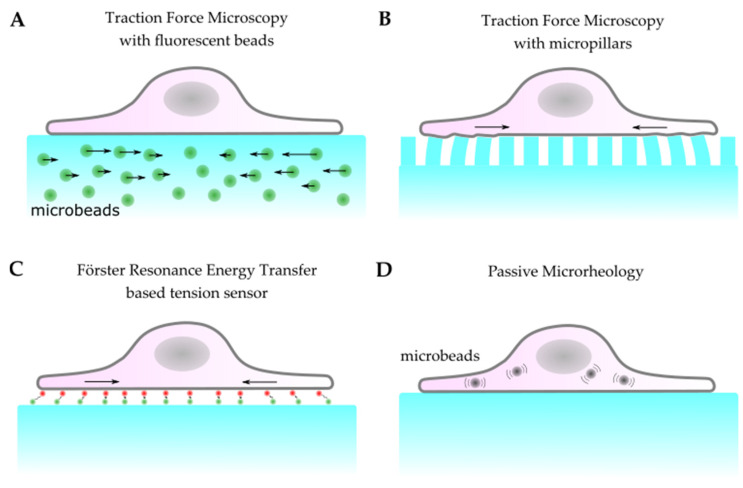
Schematics of passive methods to measure cellular forces and mechanical properties. (**A**) Embedding fluorescent beads in substrates with a known compliance allows for the mapping of cellular traction forces. (**B**) The deformation of micropillars by cells seeded on top of them can be used to calculate discrete cellular traction forces. (**C**) Specifically engineered molecular tension sensors based on Förster-Resonance-Energy-Transfer can be utilized to measure forces at cell-substrate contacts. (**D**) In passive micro-rheology, the movement of beads taken up by cells, which is caused by Brownian motion, allows for the study of the viscoelastic properties of cells.

**Figure 5 life-11-00691-f005:**
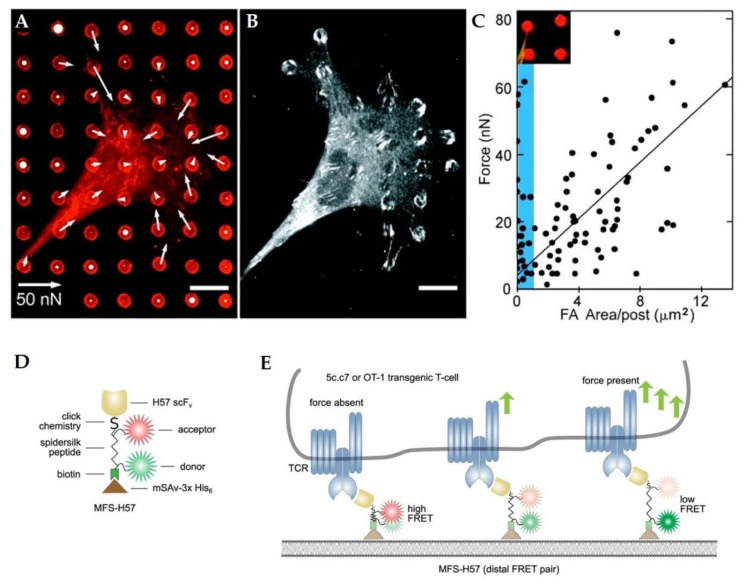
Passive cellular force sensors. (**A**) Confocal image of a smooth muscle cell on a micropillar substrate. Fibronectin staining (red) allows for the calculation of cellular forces (white arrows.). (**B**) Immunofluorescence staining of vinculin for the calculation of focal adhesion areas. (**C**) The exerted forces on each pillar in relation to the focal adhesion area per post. (Inset) Deflection of a post due to cellular force. (**D**) Illustration of a Förster-Resonance-Energy-Transfer (FRET)-based force sensor. (**E**) The exerted force leads to a stretching of the sensor, which results in a reduced FRET efficiency. (**A**–**C**) were adapted with permission from [174]. Copyright (2003) National Academy of Sciences, U.S.A. (**D**–**E**) were adapted from.

## Data Availability

Not applicable.

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
