# Peer review of "In Vitro Measurements of Cellular Forces and their Importance in the Lung—From the Sub- to the Multicellular Scale"

_life, 2021, doi:10.3390/life11070691_

Round 1

Reviewer 1 Report

This contribution by Kolb et al. provides a well written and extensively referenced overview of vast range of experimental methods that are currently available for measuring cellular forces at length scales ranging from subcellular to whole organs.  In general, it is easy to read and many basic concepts are well explained, but at times too trivial for readers of scientific literature.   

Major suggestion:

Example data/images for some methods that are described would greatly enrich this review. Especially for readers from outside the field reading plain text without seeing examples of the actual force measurements data could be frustrating.  

Minor editing suggestions:

Ln. 55: "Overall, a soft body is deformed more by an applied stress than a hard body." 

This sentence is too trivial, can by omitted.

“In addition to applying strain by pulling cells, mechanics can also be studied.” This sentence is too trivial, can by omitted.

Ln. 159: "In addition to applying strain by pulling cells, mechanics can also be studied ...."

Rather ”applying stress” or “causing/inducing strain” should be used.

Ln. 191-2 ...the necessary forces to push particles out of the trap center can be acquired.  

Measured (or estimated) would be a better word?

Ln 255. TFM abbreviation not defined – should be introduced at first mentioning of traction force microscopy ln. 241.

Reviewer 2 Report

The article by Kolb, P. et al. about the mechanical forces relevant at a cellular level in the lung and the mechanisms to measure them appears scientifically correct, well organised and counts with extense bibliography supporting their affirmations, but the text needs a significant language revision. There are not many complicated sentences, but the text have various strange expressions and constructions that difficult the reading. Moreover many words seems erroneously chosen and sometimes there are inconsistences regarding verb tense and singular/plural form concordance. Some examples are mentioned as minor comments, but the text is plenty of minor faults like the mentioned.

Major comments:

- The title “Measuring Cellular Forces in vitro – from the Sub- to the Multi-2 cellular Scale” gives a partial idea about the subject of review. It´s true that the current title indicates that is centered on the possibilities to measure forces in vitro (which appears correct), but do not mention the fact that the work is limited to the lungs. The current form is hyperbolic and induces misunderstanding, as long as the real subjet are the methods employed to study mechanical forces at a cellular level in the lung.

- Abstract. I think it is difficult to read due to some language issues, but the main problem is that it seems like a justification of the performed work instead of an abstract or summary of the main text.

Minor comments:

- 1. Introduction. Lines 24-26. You can merge the two sentences to be less reiterative.

- 1. Introduction. Line 26. You can simplify: “comprise stretching, compressing and shear stress”.

- 1.Indroduction. Line 27. You tell understanding twice in 10 words.

- 2. Forces on Lung. It is repeated the word “boundary”. I do not feel it is correct. This word refers to an edge or a contour of something. If you use this word you have to specify what is that “something”. There is probably a better word to reflect author´s intention.

- 3. Cellular force measurements. Lines 127-9. The first sentence is complex and I think it can be easily simplified removing some parts to facilitate the reading.

- 3.1. Line 234. You should mention in the text the cited ion channel (Piezo1, or just Piezo channels).

Round 2

Reviewer 2 Report

The article by Kolb, P. et al. about the mechanical forces relevant at a cellular level in the lung and the mechanisms to measure them has been notably improved. Both major comments were addressed, correctly in my opinion. However there are still some minor faults mentioned in the next lines:

- 2. Forces on Lung. Line 110. I think the correct word is “plays”, instead of “pays”.

- 2. Forces on Lung. Line 128. “The stiffness of the is defined” seems wrong, you may include the word “matrix” or remove “of the”.

- 3. Cellular force measurements. Line 161. It looks better “In recent years, a” than “In recent year, a”.

- 3. Cellular force measurements. Figure 1 legend, line 180. Please remove “and” after the comma.

- 3.2. Line 338. You should remove “the” before “precise calculation of exerted”. You could also include a comma “,” in this sentence.

- 3.2. Line 390. You should change “This can be used to created” for “This can be used to create”.

- 3.2. Line 402. You should remove “that” after “microparticles”.

- 4. Line 448. Please joint “tech nologies”.
